# Nutritional status positively impacts humoral immunity against its *Mycobacterium tuberculosis*, disease progression, and vaccine development

Mamiko Niki[1], Takashi Yoshiyama[2], Hideaki Nagai[3], Yuji Miyamoto[4], Makoto Niki[1], Ken-ichi Oinuma[1], Taishi Tsubouchi[1], Yukihiro Kaneko[1], Sohkichi Matsumoto[5], Yuka Sasaki[2], Yoshihiko Hoshino[4]*

1 Department of Bacteriology, Osaka City University Graduate School of Medicine, Abeno, Osaka, Japan, 2 Fukujuji Hospital, Japan Anti-Tuberculosis Association, Matsuyama, Kiyose, Tokyo, Japan, 3 National Hospital Organization, Tokyo National Hospital, Takeoka, Kiyose, Tokyo, Japan, 4 Department of Mycobacteriology, Leprosy Research Center, National Institute of Infectious Diseases, Aoba, Higashi-Murayama, Tokyo, Japan, 5 Department of Infectious Disease Control and International Medicine, Niigata University Graduate School of Medical and Dental Sciences, Niigata, Japan

* yhoshino@nih.go.jp

## Abstract

Nutritional status contributes to the regulation of immune responses against pathogens, and malnutrition has been considered as a risk factor for tuberculosis (TB). *Mycobacterium tuberculosis* (Mtb), the causative agent of TB, can modulate host lipid metabolism and induce lipid accumulation in macrophages, where the bacilli adopt a dormant phenotype. In addition, serum lipid components play dual roles in the regulation of and protection from Mtb infection. We analyzed the relationship between nutritional status and the humoral immune response in TB patients. We found that serum HDL levels are positively correlated with the serum IgA specific for Mtb antigens. Analysis of the relationship between serum nutritional parameters and clinical parameters in TB patients showed that serum albumin and CRP levels were negatively correlated before treatment. We also observed reduced serum LDL levels in TB patients following treatment. These findings may provide insight into the role of serum lipids in host immune responses against Mtb infection. Furthermore, improving the nutritional status may enhance vaccination efficacy.

## Introduction

Tuberculosis (TB) is one of the leading causes of death worldwide and remains a major global public health problem. According to the WHO Global Tuberculosis Report 2017, about 10 million people newly developed TB and 1.6 million died [1]. In addition, it was reported that 23% of the world population is latently infected with *Mycobacterium tuberculosis* (Mtb), the causative pathogen of TB, and is at high risk of Mtb reactivation.

**Funding:** The authors are thankful to all healthcare workers and tuberculosis patients who participated in this study. This work was supported in part by AMED under Grant Number JP18FK0108075 (to Y. H.) and by a Grant-in- Aid for Scientific Research (C) from the Japan Society for the Promotion of Science for Mam. N. (16K09584) and Y. H. (18K08312). The funders had no role in study design, data collection and analysis, decision to publish, or preparation of the manuscript.

**Competing interests:** The authors have declared that no competing interests exist.

As the nutritional status of the host plays a significant role in the maintenance of health status, the relationship between nutrition and infectious diseases has been widely studied [2–4]. In TB studies, the link of malnutrition and disease progression has long been recognized [5], and the TB incidence rate is higher in developing countries where malnutrition and bad hygiene deteriorate health. The relationship between nutrition and TB was examined using animal models in the mid-20th century [6]. Harries et al. reported that TB patients have a higher degree of under-nutrition compared to healthy hospital staffs [7]. Another study indicated that patients with under-nutrition at the time of TB diagnosis showed a 2-fold higher risk of death [8]. Although researches for many years have demonstrated the impact of nutrition on the progression and reactivation of TB, the association between host lipid profile and TB remains controversial. Some epidemiological studies showed that obesity is protective against TB [9, 10], while Mtb preferentially acquires host lipids in order to cause and maintain disease [11], and dietary cholesterol intake positively correlates with an increased risk of TB progression [12].

Mtb infection forms granuloma comprised of fibroblasts that surround Mtb-infected macrophages. Whole genome analysis revealed that progression of the infection within granuloma is accompanied by dysregulation of host lipid metabolism [13]. Mtb induces the differentiation of macrophages to lipid-loaded foamy cells, and Mtb acquires a dormancy-like phenotype [14, 15]. An *in vitro* study showed that *Mycobacterium avium* cells in lipid-laden macrophages cease division, and the withdrawal of lipids from the culture medium triggers the decline of lipid bodies in macrophages and replication of the bacilli [16]. These data indicate that lipid profile changes in hosts may affect the Mtb cellular metabolic status in macrophages and overall disease status.

As Mtb is an intracellular pathogen, the activation of cell-mediated immunity has long been believed as crucial for protection from TB [17]. However, accumulating experimental evidence suggests that humoral immunity can modulate the immune response to intracellular pathogens [18–20], and that humoral immunity acts as an important component of protective immune responses to Mtb [21]. Current research indicates that IgG antibodies obtained from TB patients promoted Mtb infection in alveolar epithelial cells, whereas IgA antibodies inhibited Mtb infection [22]. Other research showed that purified IgA obtained from human colostrum exhibited an inhibitory effect against Mtb infection in mice [23]. In addition, IgA-deficient mice showed increased susceptibility against *Mycobacterium bovis* infection [24]. In addition to offering some measure of protection, recent studies have highlighted the potential of serum IgA levels as a marker of Mtb infection. It has been reported that elevated serum IgA levels against the Mtb antigen lipoprotein Z are observed in individuals with latent TB infection [25]. IgA antibodies targeting other Mtb antigens such as ESAT-6, CFP10, Rv2031(Acr) [26] and lipoarabinomannan [27] are also detected in Mtb-infected patients. These findings suggest that serum IgA acts as an immunological marker of Mtb infection in the airway mucosa as well as a marker for the induction of mucosal immunity.

Based on these findings, many researchers have focused on the induction of humoral immunity by developing a new TB vaccine to replace BCG, whereas others are focusing on improving the efficacy of BCG. The phase 2b trial of M72/AS01$_E$ noted the induction of humoral immunity as well as cellular immunity and enhanced vaccine effectiveness [28]. On the other hand, several studies attempted to induce B cell responses against BCG by mucosal vaccination [29–31]. The importance of humoral immunity in TB vaccine development has been well reviewed elsewhere [21, 32, 33].

We previously evaluated the association between clinical status and serum antibody levels against several Mtb antigens in order to assess the potential of different antigens as novel vaccine candidates that elicit humoral immunity, and found that serum levels of Mtb antigen-

specific IgA, not IgG, correlated with the positive clinical statuses of TB patients [34]. In this study, we analyzed whether the patients' nutritional statuses affected the induction of specific isotypes of serum antibodies against Mtb. We also examined the relationship between serum nutritional parameters and clinical parameters in TB patients before and after treatment.

## Subjects and methods

### Participants

Patients of Fukujuji Hospital, and Tokyo Hospital, Tokyo Japan were consecutively enrolled after giving written informed consent. Patients were diagnosed with active-phase tuberculosis based on clinical symptoms, chest X-ray images, and bacterial cultures. When blood samples both before and after treatment were available, they were included in the analysis. A total of 22 patients in Fukujuji Hospital (age; 57.3 ± 14.8 yrs, males; 15, females; 7) were analyzed. All patients took Japanese standard medications for tuberculosis [35]. The following information was obtained from all patients at the time of enrollment: history of prior TB disease, work history in any healthcare setting or recent exposure to a patient with active TB, and other TB risk factors, such as having immunodeficiency disorders or taking immunosuppressive drugs. We used the same inclusion/exclusion criteria as in a previous study [34]. Information on previous medical history and clinical signs and symptoms were also collected as previously described. "The Japanese Society for Tuberculosis Classification" (1959) was applied to measure the severity of chest radiography at entry [36]. Briefly, tuberculosis lesions were classified by chest X-ray findings as type (cavity) and extent. "X-ray type (cavity)" was sub-divided from III to I (III: no cavity, II: morbid foci other than I, I: widespread cavities) and "X-ray extent" from 1 to 3 (1: minimal, 2: moderate, 3: severe). "Smear at entry" (entry = point of diagnosis before treatment) indicates the number of acid-fast bacilli inspected in the sputum smear taken at entry. The severity was subdivided as 0 (no acid fast bacilli (AFB) on smear), ± (1–2 AFB per 300 field), 1+ (1–9 AFB per 100 field), 2+ (more than 10 AFB per 100 field), and 3+ (more than 10 AFB per field). Blood sample collection was performed before treatment and after treatment. Serum concentrations of C-reactive protein (CRP), albumin, high-density lipoprotein (HDL), low-density lipoprotein (LDL) and total cholesterol were measured using a TBA-2000FR chemical analyzer (Toshiba Medical Systems Corporation, Tochigi, Japan) in the Division of Hematology in Niigata University Hospital. The research protocol was approved by the Institutional Review Boards of Osaka City University Graduate School of Medicine, Osaka, Japan, Tokyo Hospital, Tokyo, Japan, Fukujuji Hospital, Tokyo, Japan, and by the Research Ethics Committee of the National Institute of Infectious Disease, Tokyo, Japan.

### Measurement of serum antibody levels

Concentrations of IgG and IgA antibodies specific for Mtb were determined by ELISA using recombinant proteins as previously described with slight modification [34]. Ninety-six well microplates (Sumilon Type H, LMS, Tokyo, Japan) were coated with each recombinant antigen in bicarbonate buffer, pH 9.6 and were blocked with phosphate buffered saline (PBS) containing 0.05% Tween 20 and 5% skim milk. Human serum samples diluted 1:200 in PBS containing 0.05% Tween 20 and 0.5% skim milk were added in duplicate (IgG) or triplicate (IgA) to the antigen-coated wells. HRP-conjugated anti-human IgG or IgA antibodies were added at a 1:2000 or 1:1000 dilution, respectively. For the visualization of the reactions, 100 μL of SureBlue reserve-TMB was added to each well. The reactions were stopped by acidification and the absorbance of each reaction solution was measured at 450 nm using a Multiskan Spectrophotometer (Thermo Fisher Scientific, Yokohama, Japan). The results of the IgG-ELISA were expressed as absorbance at 450 nm, whereas results of the IgA-ELISA were expressed as

ELISA-Index, S / (B+3SD), where S is the average OD value of the duplicate test samples and B +3SD corresponds to the average OD value of the duplicate negative controls (B) plus three times the standard deviation (SD) [26].

## Reagents and recombinant protein preparation

pET-21b, pET-22b, Luria-Bertani (LB) medium and carbenicillin were obtained from Sigma (St. Louis, MO, USA); isopropyl-1-thio-beta-D- galactopyranoside and Ni-NTA agarose were obtained from Qiagen (Gaithersburg, MD, USA); skim milk was from Morinaga (Tokyo, Japan); horseradish peroxidase-conjugated anti-human IgG or IgA antibodies was from Dako (Carpinteria, CA, USA); and SureBlue reserve TMB microwell peroxidase substrate was obtained from KPL (Gaithersburg, MD, USA). Expression and purification procedures for recombinant mycobacterial antigens (ESAT-6, CFP10, MDP1, Ag85A, Acr, HBHA and HrpA) were described previously [34].

## Statistical analysis

The Mann-Whitney U-test was used to compare IgG and IgA levels between two independent groups, whereas one-way ANOVA was used for the comparison of three or more unmatched groups. Spearman's rank correlation coefficient was used to determine the correlation of two independent values among ELISA values, nutritional status values and the severity of clinical status values. Wilcoxon signed-rank test was used for comparison of before-after treatment data in TB patients. All analyses were performed using online statistics calculators (http://www.socscistatistics.com/tests/Default.aspx, http://vassarstats.net/index.html). The threshold of significance was set at $p < 0.05$.

## Results

### Changes of antibody levels during treatment

We evaluated whether TB treatment affects the serum antibody levels for various Mtb antigens among TB patients. Previously, we observed a significant decrease in IgG levels against Acr and HrpA [37]. In this study, we could not find any changes in antibody levels during treatment (Figs 1 and 2).

### Clinical parameters and antibody levels

In order to clarify the relationship between serum antibody quantity and the disease status, serum antibody levels were compared to clinical parameters. Consistent with our previous observation [37], IgG levels against HrpA before treatment showed a positive correlation with "Smear at entry" (Fig 3).

### CRP and serum nutritional parameters

We analyzed the association between serum CRP and serum nutritional parameters (HDL, LDL, total cholesterol and albumin) at the onset of treatment. We found that the serum CRP levels were negatively correlated with serum albumin levels (Fig 4A). Although we could not find a direct correlation between CRP and lipid parameters, analysis of the relationship among nutritional parameters indicated that serum levels of HDL, LDL and total cholesterol were found to have positive correlations with serum albumin levels in TB patients before treatment (Fig 4B). Interestingly, analysis of serum samples obtained after treatment revealed that all nutritional parameters tested in this study showed negative correlations with CRP (Fig 4C),

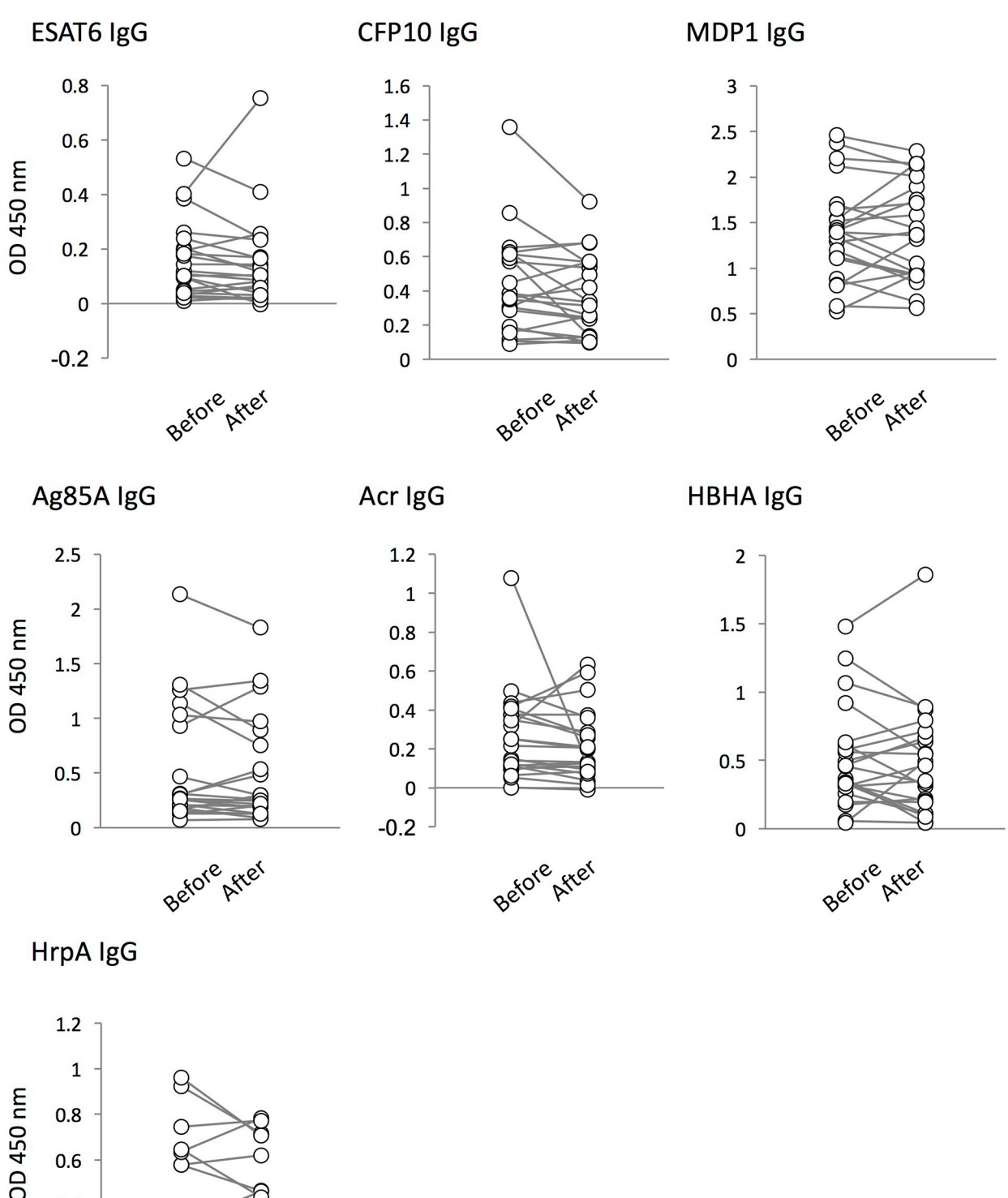

**Fig 1. Comparison of IgG responses to Mtb antigens in active TB (ATB) patients before and after treatment.** The levels of serum IgG in ATB patients against 7 antigens (ESAT-6, CFP-10, MDP1, Ag85A, Acr, HBHA, and HrpA) before treatment (labeled as "Before") and after treatment ("After") were analyzed by ELISA. Data shown are the average of triplicate experiments. There were no statistical changes in the levels of serum IgG against TB antigens before and after treatment.

and serum albumin level were positively correlated with LDL and total cholesterol levels (Fig 4D).

## Disease severity and serum parameters

We observed whether disease severity diagnosed by chest X-ray and sputum smear indexes was associated with serum parameters including serum CRP. As we previously reported [34], significant associations were seen between serum CRP levels before treatment and "X-ray extent" as well as "Smear at entry" indexes (S1A Fig). In addition, serum albumin levels showed negative relationships with "X-ray extent" and "Smear at entry" indexes (S2A Fig). However, we could not find any association between disease severity and lipid parameters. We also evaluated whether the serum parameters after treatment correlated with clinical status at the onset of treatment. As expected, serum CRP levels showed a positive correlation with disease status. Additionally, we found a similar relationship between serum albumin levels and disease status (S1B and S2B Figs). On the other hand, there was no relationship between lipid parameters after treatment and disease status.

## Correlation between antibody levels and nutritional status before and after treatment

In order to clarify whether nutritional status affected the host immune response, antibody levels were compared to nutritional parameters. Surprisingly, serum levels of HDL and total cholesterol showed positive relationships with serum IgA levels against Mtb antigens before treatment (Fig 5A and 5B). On the other hand, no correlations were observed between serum antibody levels and serum nutritional parameters after treatment.

## Change in serum nutritional status during treatment

Finally, we analyzed whether TB treatment affected the serum levels of nutritional parameters. Although we could not find any differences in levels of serum HDL, total cholesterol and albumin during treatment, a significant decrease was observed in serum LDL levels after treatment (Fig 6).

## Discussion

In this study, we evaluated the relationship between immune responses against Mtb infection and host nutritional status. It has been reported that elevated CRP levels and decreased albumin levels are often seen in many inflammatory diseases, and are correlated with disease severity and mortality rate [38–41]. Artero et al. reported that hypoalbuminemia is a risk factor for mortality in patients with community-acquired bloodstream infections with severe sepsis [42]. In addition, recent studies indicated that serum albumin levels were decreased in HIV-infected TB patients [43] and were affected by antiretroviral and anti-TB therapy [44]. Consistent with these reports, we found a negative correlation between serum albumin level and CRP level in active TB patients before treatment. Although we could not find statistical significance between CRP and serum lipid levels, patients with low serum albumin levels also exhibited low serum lipid levels before treatment. Interestingly, analysis of serum samples obtained from TB

## ESAT6 IgA

## CFP10 IgA

## MDP1 IgA

## Ag85A IgA

## Acr IgA

## HBHA IgA

## HrpA IgA

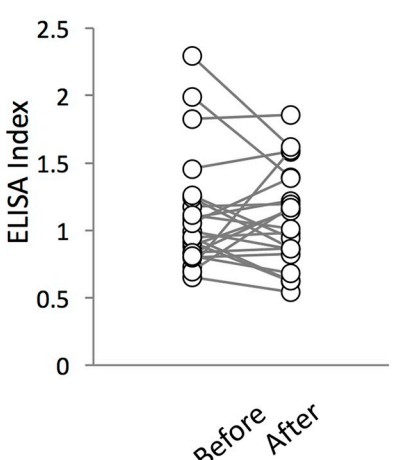

**Fig 2. IgA responses to Mtb antigens.** The levels of serum IgA in ATB patients against 7 antigens (ESAT-6, CFP-10, MDP1, Ag85A, Acr, HBHA, and HrpA) before treatment (labeled as "Before") and after treatment ("After") were analyzed by ELISA. The results of the IgA-ELISA were expressed as ELISA-Index, as described in "Subjects and Methods". Also, there were no significant differences in the levels of serum IgA against TB antigens before and after treatment.

patients after treatment revealed negative correlations between CRP and serum lipid parameters as well as serum albumin level. The role of serum lipids on inflammation during Mtb infection was not fully understand in this study because of the limitation of sample size and further study will be needed.

To clarify the role of serum nutritional status in the immune response against Mtb infection, we evaluated the correlation between serum nutritional status with Mtb-specific antibody profiles. We found a positive relationship between serum lipid parameters and the levels of Mtb antigen-specific IgA, suggesting that serum lipid components may enhance the mucosal immune response during Mtb infection. We previously reported that the presence of HDL cholesterol helps the internalization of Mtb into macrophages [45]. The spread of Mtb infection at the lung mucosa may trigger the rapid activation of the mucosal immune response, leading to elevated Mtb-specific IgA production. IgA in its secretory form is regarded as the main effector molecule of the mucosal immune system and serves as the first line of defense against pathogen invasion initiated at mucosal surfaces. One study conducted among healthcare workers with suspected latent TB infection (LTBI) patients showed that elevated serum IgA levels were seen in IFN-γ-positive subjects, suggesting a protective role of IgA in LTBI

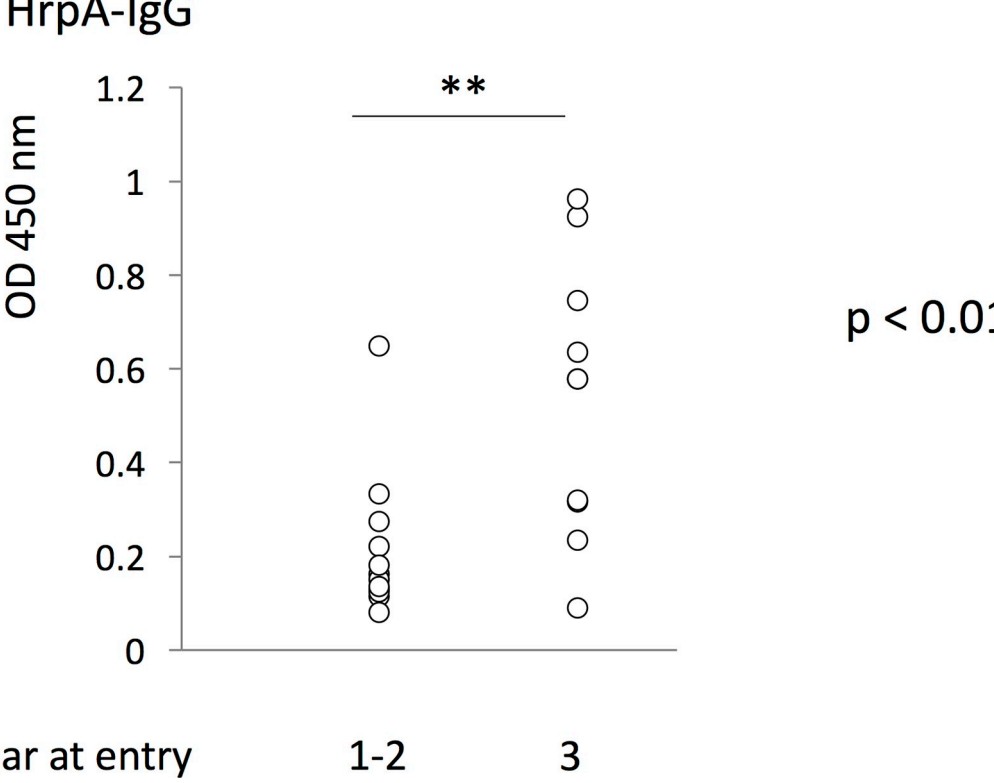

**Fig 3. The levels of serum IgG against HrpA before treatment in "Smear at entry".** The levels of serum IgG against HrpA before treatment were compared to the severity in "Smear at entry" sub-grouped between 1+, 2+ and 3+. HrpA-IgG levels before treatment were found to have a positive relationship with "smear at entry" scores. **: p < 0.01.

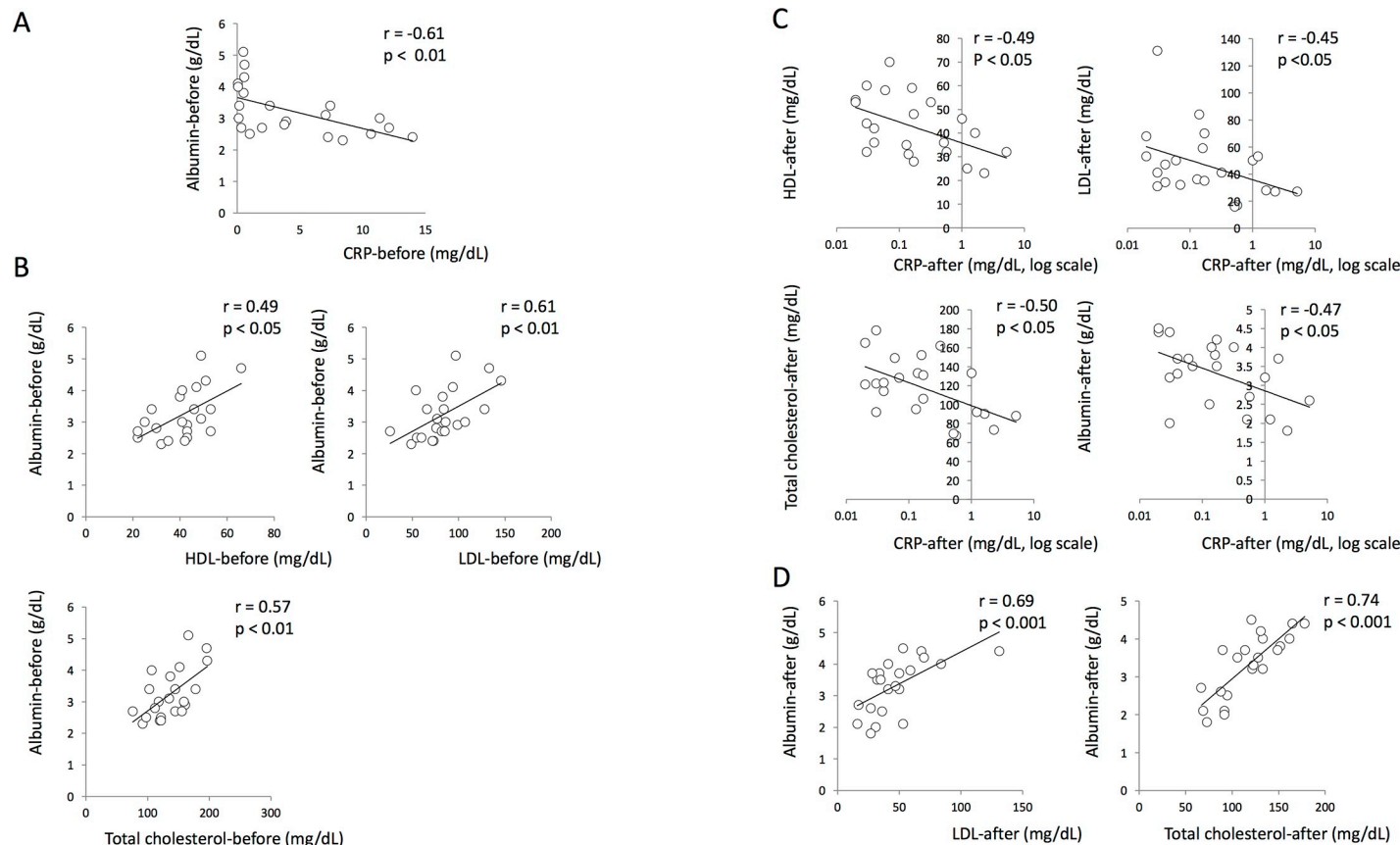

**Fig 4. Comparison of CRP and serum nutritional parameters.** (A) Correlation between serum CRP level and albumin in active TB patients before treatment. (B) The relationship between serum lipids and serum albumin levels in TB patients at the onset of treatment. (C) The relationship between serum CRP levels and nutritional parameters in TB patients after chemotherapy. (D) The relationship between serum lipid parameters and serum albumin levels in TB patients after chemotherapy.

[46]. It was also reported that IgG and IgA against mycobacterial glycolipid antigens were found to be higher in QFT-positive LTBI healthcare workers [47]. A recent study by Zimmermann et al. revealed that inhibitory activity of antibodies against Mtb infection is dependent on their isotype. They found that purified serum IgG obtained from TB patients promoted the uptake of Mtb cells by A549 human lung epithelial cells, whereas serum IgA reduced the bacterial load. These results imply that elevated serum HDL cholesterol may be beneficial to Mtb-infected individuals for the proliferation of IgA-producing plasma cells at the early stage of infection.

Low effectiveness of BCG vaccination in elderly people has been one of the most serious problems in preventing the spread of pulmonary TB. In addition to the development of new vaccines to replace BCG, researchers have also been studying ways to enhance the efficacy of BCG. Recent studies have highlighted the importance of antibody-mediated immunity against Mtb infection [32, 48], as it is now understood that antibodies can confer protection against intracellular pathogens via Fc-receptor mediated phagocytosis [18, 49]. Therefore, improved vaccination strategies that modulate the humoral immune response are being actively studied. Since the association between nutrition and immune function has received considerable attention for many years [50, 51], nutritional status is very likely to affect vaccine efficacy [52]. It has also been reported that nutritional status affected IgA responses against various vaccinations, for example, 23 valent pneumococcal capsular polysaccharide vaccine, Pneumo-vax and

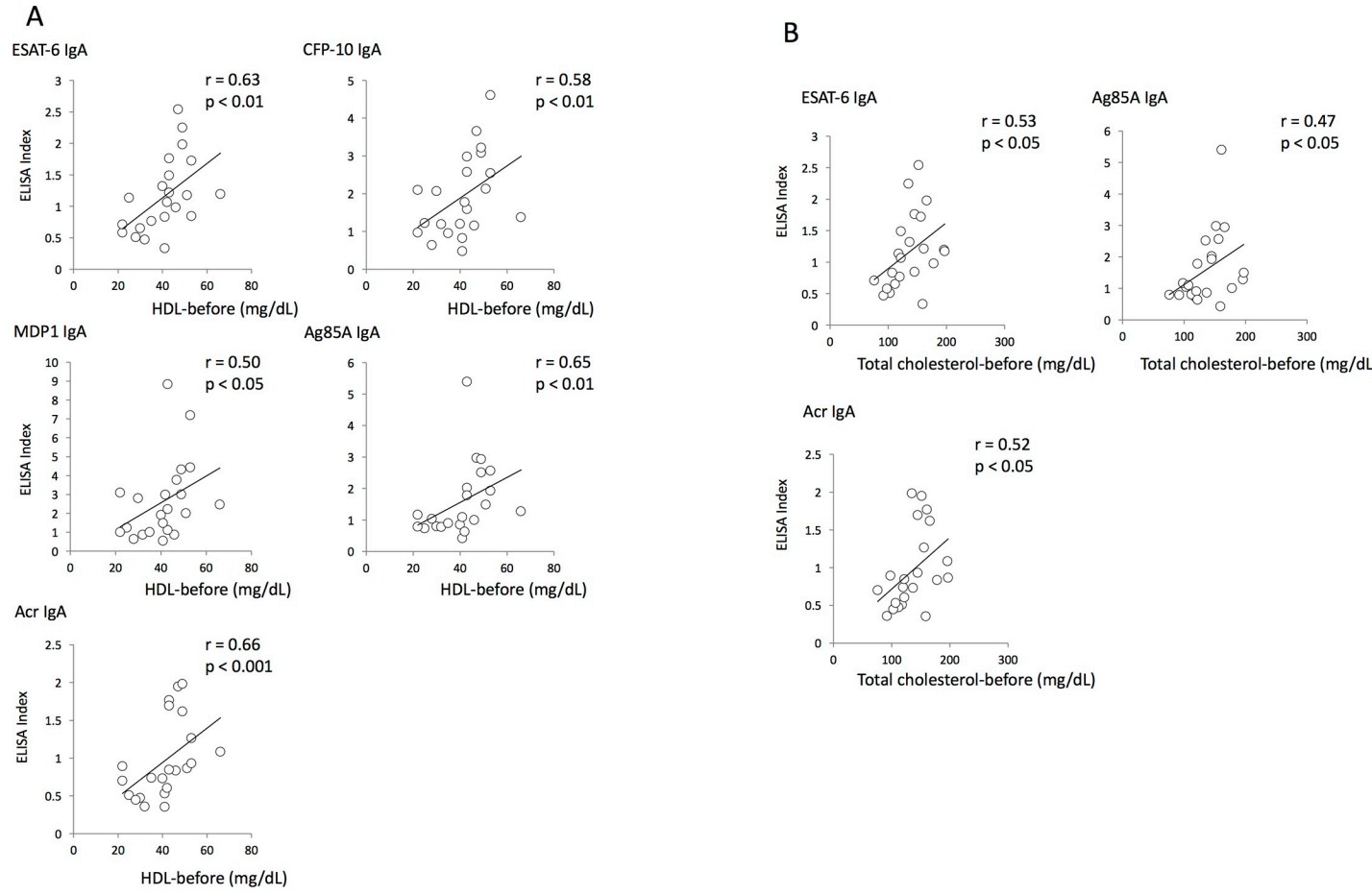

**Fig 5. Correlation between antibody levels and nutritional status before and after treatment.** (A) Correlation between serum HDL concentrations and serum IgA levels specific to Mtb antigens before TB treatment. (B) Correlation between serum total cholesterol levels and serum IgA levels specific to Mtb antigens before TB treatment.

the Human Diploid-Cell Rabies Vaccine [53–56]. Monitoring nutrition may be a practical and low-cost way to impact vaccination outcome.

Several studies indicated that low concentrations of lipids are often seen in patients with active TB, and that this correlates with disease severity [57, 58]. In addition, Akpovi et al. reported that restoration of serum HDL and total cholesterol levels was observed during treatment [59]. Contrary to the previous reports, we did not identify any changes in HDL and total cholesterol serum levels. The different results obtained in our study may be due to the nutritional status in Japan. According to the nutrition survey in 2010, dietary intake of cholesterol for Japanese adults >20 years old was in the range of 300 to 349 mg/day, which was higher than global mean dietary cholesterol intake (228 mg/day) [60]. On the other hand, we observed a significant decrease in serum LDL levels during treatment. Moreover, serum LDL levels were found to be lower in LTBI individuals than in active TB patients. It has been reported that Mtb infection induces lipid-droplet formation in macrophages [61]. The lipid-laden macrophages allow mycobacterial cells to persist and become defective in bactericidal activity [16, 61, 62]. LDL is known to be a main lipid acquisition source of foamy macrophage and the increased expression of CD36 and LOX1, macrophage surface receptors of oxidized LDL, was observed during mycobacterial infection [63, 64]. Although further studies are needed to solve the

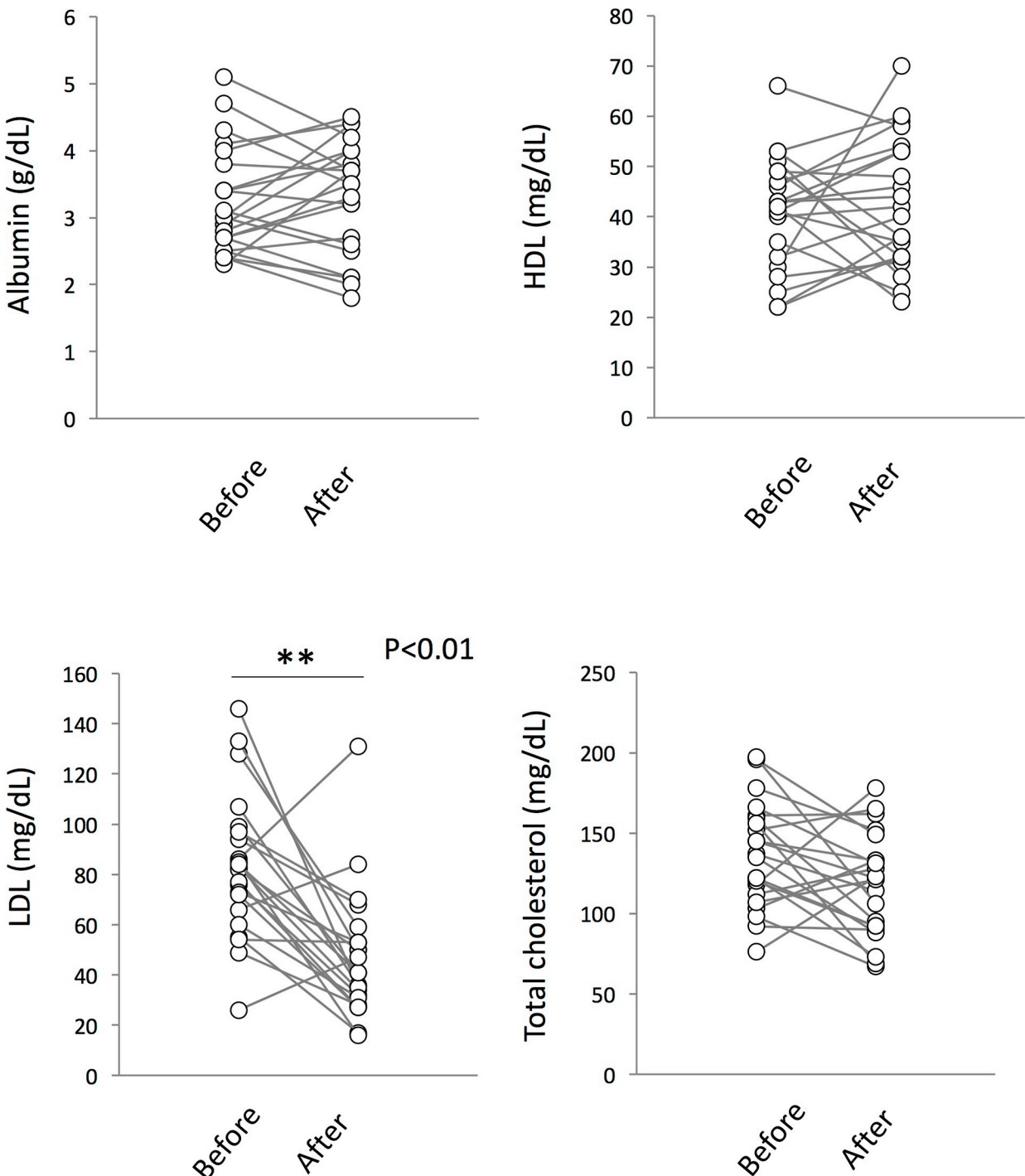

**Fig 6. Changes in serum nutritional status in ATB patients during treatment.** Analysis of serum nutritional levels showed a significant decrease in serum LDL levels in ATB patients after treatment. **: $p < 0.01$.

complicated relationship between serum lipids and TB progression, decreased LDL levels in patients after treatment may be a trigger of switching Mtb cells from an actively-growing state to a slow-growing dormant state in macrophages.

## Conclusion

To our knowledge, this is the first human study to investigate the relationship between nutritional status and humoral immunity in a Japanese population. Serum albumin levels were negatively correlated with CRP at the onset of treatment. It was also found that serum albumin levels were positively associated with serum lipid levels. Evaluation of serum antibodies and nutritional parameters revealed that enhanced serum IgA specific to Mtb antigens was observed in TB patients with high serum HDL levels before TB treatment. On the contrary, serum LDL levels were decreased in both TB patients after treatment and LTBI individuals. The data in this study may provide an insight into the role of serum lipids in host immune responses against Mtb infection as well as in the acquisition of a dormancy-like phenotype by Mtb cells. The control and maintenance of optimal nutritional status may improve the outcome of vaccination against TB.

## Supporting information

**S1 Fig. Association between serum CRP levels and disease severity.** (A) Correlation between serum CRP before treatment and disease severity at onset. (B) Correlation between serum CRP after treatment and disease severity at onset. For statistical analyses, groups I and II of "X-ray cavity" and groups 1 and 2 of "Smear at entry are combined because of the small sample number of patients diagnosed as group I in "X-ray cavity" and 1 in "Smear at entry". **: $p < 0.01$, *: $p < 0.05$.
(TIFF)

**S2 Fig. Association between serum albumin levels and disease severity.** (A) Correlation between serum albumin before treatment and disease severity at onset. (B) Correlation between serum albumin after treatment and disease severity at onset. **: $p < 0.01$, *: $p < 0.05$.
(TIFF)

## Author Contributions

**Conceptualization:** Mamiko Niki, Yoshihiko Hoshino.

**Data curation:** Mamiko Niki, Yoshihiko Hoshino.

**Formal analysis:** Mamiko Niki.

**Funding acquisition:** Mamiko Niki, Yoshihiko Hoshino.

**Investigation:** Mamiko Niki, Sohkichi Matsumoto, Yoshihiko Hoshino.

**Methodology:** Mamiko Niki.

**Project administration:** Mamiko Niki, Yoshihiko Hoshino.

**Resources:** Takashi Yoshiyama, Hideaki Nagai, Yuji Miyamoto, Makoto Niki, Ken-ichi Oinuma, Taishi Tsubouchi, Yukihiro Kaneko, Sohkichi Matsumoto, Yuka Sasaki.

**Software:** Mamiko Niki.

**Supervision:** Yoshihiko Hoshino.

**Writing – original draft:** Yoshihiko Hoshino.

**Writing – review & editing:** Yoshihiko Hoshino.

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
