## [Decision Letter · Decision Letter 0]

12 Nov 2019

PONE-D-19-26473

Nutritional status positively impacts humoral immunity against its Mycobacterium tuberculosis, disease progression, and vaccine development.

PLOS ONE

Dear Dr. Hoshino,

Thank you for submitting your manuscript to PLOS ONE. After careful consideration, we feel that it has merit but does not fully meet PLOS ONE’s publication criteria as it currently stands. Therefore, we invite you to submit a revised version of the manuscript that addresses the points raised during the review process.

We would appreciate receiving your revised manuscript. To enhance the reproducibility of your results, we recommend that if applicable you deposit your laboratory protocols in protocols.io, where a protocol can be assigned its own identifier (DOI) such that it can be cited independently in the future. For instructions see: http://journals.plos.org/plosone/s/submission-guidelines#loc-laboratory-protocols

We look forward to receiving your revised manuscript.

Kind regards,

Frederick Quinn

Academic Editor

PLOS ONE

Journal Requirements:

1. Please provide additional details regarding participant consent. In the ethics statement in the Methods and online submission information, please ensure that you have specified (1) whether consent was informed and (2) what type you obtained (for instance, written or verbal, and if verbal, how it was documented and witnessed). If your study included minors, state whether you obtained consent from parents or guardians. If the need for consent was waived by the ethics committee, please include this information

Reviewers' comments:

Reviewer's Responses to Questions

**Comments to the Author**

1. Is the manuscript technically sound, and do the data support the conclusions?

Reviewer #1: Yes

Reviewer #2: Partly

2. Has the statistical analysis been performed appropriately and rigorously? 

Reviewer #1: I Don't Know

Reviewer #2: No

3. Have the authors made all data underlying the findings in their manuscript fully available?

Reviewer #1: Yes

Reviewer #2: Yes

4. Is the manuscript presented in an intelligible fashion and written in standard English?

Reviewer #1: Yes

Reviewer #2: Yes

5. Review Comments to the Author

Reviewer #1: Thank you for submitting the manuscript. There are few questions/ comments that will help clarify few things

1. Please provide a reason not to include nutritional data from patients used in your earlier published studies. This would increase the number of patients and benefit the statistical analysis.

2. Please provide more demographic details about the 22 patients enrolled in this study

3. Please consider either adding more LTBI patients or removing those patients from the study. Only 6 data points is not enough to come to any conclusion.

4. IS there any correlation between serum albumin or HDL versus TNFa levels in active TB patients?

5. Please provide Fig 5B

Reviewer #2: Firstly and most importantly, the topic is too big for a small scale of study. As stated, 22 patients and 6 patients from 2 hospital, may not be sufficiently supportive for the correlation study. I also doubt about the statistical analysis for such small number of patients. I strongly recommend the authors to perform the study on a larger scale, on the number of both medical centers and patients.

Secondly, a lot of the results and conclusions are based on only observational and descriptive data. For example, it’s pretty easy to perform such study, by describing what data and thus correlation from such results. However, how to further confirm and verify, still remains unclear in this study. For example, could the authors provide evidence on the correlation as concluded, by changing the lipid level then obtaining the change in TB development?

Thirdly, one slight drawback is, since the authors had stated the nutritional status, however, as seen from the section of Materials and Methods, the authors only stated all patients will follow the standard treatment in Japan for TB. Then how about the diets, which will eventually change their nutritional status? Consequently, detailed description are required to support the results and conclusions.

6. PLOS authors have the option to publish the peer review history of their article (what does this mean?). If published, this will include your full peer review and any attached files.

Reviewer #1: No

Reviewer #2: No

---

## [Author Response · Author response to Decision Letter 0]

7 May 2020

30th April, 2020

Professor Fredrick Quinn, 

Academic Editor of PLOS ONE

Re: PONE-D-19-26473

Dear Prof. Fredrick Quinn,

We appreciate the opportunity to revise our original manuscript entitled “Nutritional status positively impacts humoral immunity against its Mycobacterium tuberculosis, disease progression, and vaccine development.”. The original manuscript has been dramatically modified according to the reviewers’ comments and suggestions, and we believe the revised manuscript will be more informative for readers of your journal. 

Our responses to the reviewers’ comments are addressed in detail below. 

Again, we are grateful for the opportunity to submit a manuscript for publication in PLOS ONE.

Sincerely Yours, 

Yoshihiko Hoshino, MD, PhD

Department of Mycobacteriology National Institute of Infectious Diseases, Higashi-Murayama, Tokyo, JAPAN 

We wish to thank the reviewer for the insightful comments and suggestions on our paper. The comments have helped us improve the paper. 

1. Please provide additional details regarding participant consent. In the ethics statement in the Methods and online submission information, please ensure that you have specified (1) whether consent was informed and (2) what type you obtained (for instance, written or verbal, and if verbal, how it was documented and witnessed). If your study included minors, state whether you obtained consent from parents or guardians. If the need for consent was waived by the ethics committee, please include this information. 

Informed consent was obtained as a written form. We added the information in line 101, page 6.

In accordance to the reviewer #1’s suggestion, we have removed the data obtained from LTBI patients and the phrase “data not shown” that refers to the data corresponding to the serum nutritional levels of LTBI patients.

Answers to the comments

Reviewer #1

We strongly appreciate the reviewer’s comments and suggestions on this paper.

Answers to the specific comments:

1. Please provide a reason not to include nutritional data from patients used in your earlier published studies. This would increase the number of patients and benefit the statistical analysis.

As the blood sample volumes obtained from the patients were low and limited, we could not afford to determine the serum biochemical parameters in addition to the analysis of serum antibody levels. In this study, we collected only the samples with enough volume for the measurement of serum nutritional status and the antibody levels at the same time.

2. Please provide more demographic details about the 22 patients enrolled in this study. 

In accordance to the reviewer’s comment, we have added the demographic details about the patients to the line 104, page 6.

3. Please consider either adding more LTBI patients or removing those patients from the study. Only 6 data points is not enough to come to any conclusion. 

According to the reviewer’s comment, we have removed LTBI patients from this study.

4. IS there any correlation between serum albumin or HDL versus TNFa levels in active TB patients? 

We agree to make an analysis between HDL and inflammatory cytokines such as TNFa. Unfortunately, we have not determined the serum TNF-alpha levels in this study. As stated in 1, we only collected small volume of samples.

5. Please provide Fig 5B.

We are sorry not including Fig 5B now we have attached Fig 5B in the file.

Reviewer #2

Firstly and most importantly, the topic is too big for a small scale of study. As stated, 22 patients and 6 patients from 2 hospital, may not be sufficiently supportive for the correlation study. I also doubt about the statistical analysis for such small number of patients. I strongly recommend the authors to perform the study on a larger scale, on the number of both medical centers and patients.

We agree to the reviewer #2. If we could expand the numbers however, we are afraid we could not, which is a limitation of this study.

Secondly, a lot of the results and conclusions are based on only observational and descriptive data. For example, it’s pretty easy to perform such study, by describing what data and thus correlation from such results. However, how to further confirm and verify, still remains unclear in this study. For example, could the authors provide evidence on the correlation as concluded, by changing the lipid level then obtaining the change in TB development?

We really agree to the reviewer. However, this is just observational and descriptive study. Further study and analysis should be required, which we completely agree.

Thirdly, one slight drawback is, since the authors had stated the nutritional status, however, as seen from the section of Materials and Methods, the authors only stated all patients will follow the standard treatment in Japan for TB. Then how about the diets, which will eventually change their nutritional status? Consequently, detailed description are required to support the results and conclusions. 

As far as diets, we did not evaluate these items. All patients were given regular meals. No special diets were considered.

---

## [Decision Letter · Decision Letter 1]

4 Jun 2020

PONE-D-19-26473R1

Nutritional status positively impacts humoral immunity against its Mycobacterium tuberculosis, disease progression, and vaccine development.

PLOS ONE

Dear Dr.Hoshino,

Thank you for submitting your manuscript to PLOS ONE. After careful consideration, we feel that it has merit but does not fully meet PLOS ONE’s publication criteria as it currently stands. Therefore, we invite you to submit a revised version of the manuscript that addresses the points raised during the review process.

Please submit your revised manuscript. If you will need more time than this to complete your revisions, please reply to this message or contact the journal office at plosone@plos.org. Please include the following items when submitting your revised manuscript:

We look forward to receiving your revised manuscript.

Kind regards,

Frederick Quinn

Academic Editor

PLOS ONE

Reviewers' comments:

Reviewer's Responses to Questions

**Comments to the Author**

1. If the authors have adequately addressed your comments raised in a previous round of review and you feel that this manuscript is now acceptable for publication, you may indicate that here to bypass the “Comments to the Author” section, enter your conflict of interest statement in the “Confidential to Editor” section, and submit your "Accept" recommendation.

Reviewer #1: (No Response)

Reviewer #2: (No Response)

2. Is the manuscript technically sound, and do the data support the conclusions?

Reviewer #1: Partly

Reviewer #2: Partly

3. Has the statistical analysis been performed appropriately and rigorously? 

Reviewer #1: I Don't Know

Reviewer #2: No

4. Have the authors made all data underlying the findings in their manuscript fully available?

Reviewer #1: Yes

Reviewer #2: No

5. Is the manuscript presented in an intelligible fashion and written in standard English?

Reviewer #1: Yes

Reviewer #2: Yes

6. Review Comments to the Author

Reviewer #1: Thank you for the revisions. Here are few minor changes

1. Line 52 "obesity is protective against" please provide a reference.

2. Line 104 demographics - Since gender cannot be in fractions please provide the exact number of males and females in the study.

3. Line 113 – 115 – X ray results – Is it possible to keep the scoring system consistent for X ray type and extent? Keep it either in ascending or descending order? Similarly change the figure S1? It is otherwise very confusing.

4. Line 118 and 119 – It is not clear what is +2 and +3? Is it >10 AFB per 100x field for +2? For +3, please explain more than 10 AFB per field. What is per field?

5. Line 153 and 173 please provide the details of the statistical analysis applied to " Before" and "After" samples since they are paired.

6. Figure 1 and 2 - please position the "Before" label correctly on the graph.

7. Figure 2 - What is the rationale behind showing the IgG ELISA data as absorbance 450 nm and IgA ELISA data as ELISA index? Please consider a uniform data presentation.

8. Line 177 and 178 - please remove the p values since they are not indicated in the actual graph (fig. 2).

9. For Figure 1, 2 and 3 legends - Instead of writing what was done, it would help to write the results / observations.

10. Line 177 and 188 - for statistics please write either p or P . Keep it uniform.

11. Please provide figure 4C and 4D .

12. Line 202 - change TB patents to TB patients.

13. Line 216 (Fig S1)- There is a difference between serum CRP levels before and after treatment. Just add a line stating that this is as expected. With decrease in disease severity with treatment, the serum CRP levels will decrease too.

14. Please provide Figure 5B

Reviewer #2: As seen from the response, the authors had barely made revision according to my comments. Consequently, I have to stick with the rejection decision. Thanks.

7. PLOS authors have the option to publish the peer review history of their article (what does this mean?). If published, this will include your full peer review and any attached files.

Reviewer #1: No

Reviewer #2: No

---

## [Author Response · Author response to Decision Letter 1]

7 Jul 2020

We wish to thank the reviewer for the insightful comments and suggestions on our paper. The comments have helped us improve the paper. 

Answers to the specific comments.

1. Line 52 "obesity is protective against" please provide a reference.

→In accordance to the reviewer’s comment, we have added the references to the text.

2. Line 104 demographics – Since gender cannot be in fractions please provide the exact number of males and females in the study.

→We have provided the exact number of males and females to the text.

3. Line 113 – 115 – X ray results – Is it possible to keep the scoring system consistent for X ray type and extent? Keep it either in ascending or descending order? Similarly change the figure S1? It is otherwise very confusing.

→We completely agree with a reviewer however, this scoring system was established in 1959 and was long time used as a standard scoring system in Japan. Again, we agree that the system is a little bit curious and confusing but we cannot modify the system by ourselves without any permission. 

4. Line 118 and 119 – It is not clear what is +2 and +3? Is it >10 AFB per 100x field for +2? For +3, please explain more than 10 AFB per field. What is per field?

→This is according to “diagnostic standards and classification of tuberculosis and other mycobacterial diseases (American Review of Respiratory Disease 1981, 123(3), 343–358). The scale was determined by Ziehl-Neelson staining, and per field means x1,000 field by an optical microscope (in the literature described above, there was an explanation as “Examination at 800-1000X is assumed”). 

In conversion of the conventional Gaffky scale, + accounts for G1 (conventional Gaffky scale), 1+; G2, 2+; G5, 3+; G9.

5. Line 153 and 173 please provide the details of the statistical analysis applied to "Before" and "After" samples since they are paired.

→We have added the description about statistical analysis in the text.

6. Figure 1 and 2 - please position the "Before" label correctly on the graph.

→As requested, we have corrected the position of the “Before” labels in the graphs.

7. Figure 2 - What is the rationale behindshowing the IgG ELISA data as absorbance 450 nm and IgA ELISA data as ELISA index? Please consider a uniform data presentation.

→We thank the reviewer’s comment on this point. Since we had to measure the antibody levels to multiple Mtb antigens using samples with limited volume, we evaluate IgG levels at the serum concentration as little as possible to save the samples. However, serum IgA levels are much lower than IgG when measured by the same method for the identification of serum IgG levels, we introduced the ELISA Index to evaluate the IgA levels in our previous study and the present manuscript according to the previous reports describing similar experiments. We have added the references to the text.

8. Line 177 and 178 - please remove the p values since they are not indicated in the actual graph (fig. 2).

→We have removed the p values from the legend.

9. For Figure 1, 2 and 3 legends - Instead of writing what was done, it would help to write the results / observations.

→As suggested, we have added the brief descriptions about the results to the legends.

10. Line 177 and 188 - for statistics please write either p or P. Keep it uniform.

→In accordance to the reviewer’s comment, we standardize the description of p-values in the text. Accordingly, we have also changed the “P” value to “p” in Fig. 3.

11. Please provide figure 4C and 4D.

→We appreciated this comment and corrected as the attachment.

12. Line 202 - change TB patents to TB patients.

→We have corrected the typo.

13. Line 216 (Fig S1)- There is a difference between serum CRP levels before and after treatment. Just add a line stating that this is as expected. With decrease in disease severity with treatment, the serum CRP levels will decrease too.

→In accordance to the reviewer’s suggestion, we have changed the text as follows: As expected, serum CRP levels showed a positive correlation with disease status. Additionally, we found a similar relationship between serum albumin levels and disease status (S1B and S2B Figs). On the other hand, there was no relationship between lipid parameters after treatment and disease status.

14. Please provide Figure 5B

→We appreciated this comment and corrected as the attachment.

---

## [Decision Letter · Decision Letter 2]

21 Jul 2020

Nutritional status positively impacts humoral immunity against its Mycobacterium tuberculosis, disease progression, and vaccine development.

PONE-D-19-26473R2

Dear Dr. Hoshino,

We’re pleased to inform you that your manuscript has been judged scientifically suitable for publication and will be formally accepted for publication once it meets all outstanding technical requirements.

Kind regards,

Frederick Quinn

Academic Editor

PLOS ONE

Additional Editor Comments (optional):

Reviewers' comments:

Reviewer's Responses to Questions

**Comments to the Author**

1. If the authors have adequately addressed your comments raised in a previous round of review and you feel that this manuscript is now acceptable for publication, you may indicate that here to bypass the “Comments to the Author” section, enter your conflict of interest statement in the “Confidential to Editor” section, and submit your "Accept" recommendation.

Reviewer #2: All comments have been addressed

2. Is the manuscript technically sound, and do the data support the conclusions?

Reviewer #2: Yes

3. Has the statistical analysis been performed appropriately and rigorously? 

Reviewer #2: Yes

4. Have the authors made all data underlying the findings in their manuscript fully available?

Reviewer #2: Yes

5. Is the manuscript presented in an intelligible fashion and written in standard English?

Reviewer #2: Yes

6. Review Comments to the Author

Reviewer #2: (No Response)

7. PLOS authors have the option to publish the peer review history of their article (what does this mean?). If published, this will include your full peer review and any attached files.

Reviewer #2: **Yes: **Zhenbo Xu

---

## [Editor Report · Acceptance letter]

27 Jul 2020

PONE-D-19-26473R2 

Nutritional status positively impacts humoral immunity against its Mycobacterium tuberculosis, disease progression, and vaccine development. 

Dear Dr. Hoshino:

I'm pleased to inform you that your manuscript has been deemed suitable for publication in PLOS ONE. Congratulations! Your manuscript is now with our production department. 

Kind regards, 

on behalf of

Dr. Frederick Quinn 

Academic Editor

PLOS ONE